# Acetabular Revision with McMinn Cup: Development and Application of a Patient-Specific Targeting Device

**DOI:** 10.3390/bioengineering10091095

**Published:** 2023-09-18

**Authors:** Zoltán Csernátony, Sándor Manó, Dániel Szabó, Hajnalka Soósné Horváth, Ágnes Éva Kovács, Loránd Csámer

**Affiliations:** 1Department of Orthopaedics and Traumatology, Faculty of Medicine, University of Debrecen, H-4032 Debrecen, Hungary; csz@med.unideb.hu (Z.C.);; 2Laboratory of Biomechanics, Department of Orthopaedics and Traumatology, Faculty of Medicine, University of Debrecen, H-4032 Debrecen, Hungary; manos@med.unideb.hu (S.M.);

**Keywords:** stemmed cup, hip revision surgery, patient-specific instrumentation

## Abstract

Background: Surgeries of severe periacetabular bone defects (Paprosky ≥ 2B) are a major challenge in current practice. Although solutions are available for this serious clinical problem, they all have their disadvantages as well as their advantages. An alternative method of reconstructing such extensive defects was the use of a cup with a stem to solve these revision situations. As the instrumentation offered is typically designed for scenarios where a significant bone defect is not present, our unique technique has been developed for implantation in cases where reference points are missing. Our hypothesis was that a targeting device designed based on the CT scan of a patient’s pelvis could facilitate the safe insertion of the guiding wire. Methods: Briefly, our surgical solution consists of a two-step operation. If periacetabular bone loss was found to be more significant during revision surgery, all implants were removed, and two titanium marker screws in the anterior iliac crest were percutaneously inserted. Next, by applying the metal artifact removal (MAR) algorithm, a CT scan of the pelvis was performed. Based on that, the dimensions and positioning of the cup to be inserted were determined, and a patient-specific 3D printed targeting device made of biocompatible material was created to safely insert the guidewire, which is essential to the implantation process. Results: In this study, medical, engineering, and technical tasks related to the design, the surgical technique, and experiences from 17 surgical cases between February 2018 and July 2021 are reported. There were no surgical complications in any cases. The implant had to be removed due to septic reasons (independently from the technique) in a single case, consistent with the septic statistics for this type of surgery. There was not any perforation of the linea terminalis of the pelvis due to the guiding method. The wound healing of patients was uneventful, and the implant was fixed securely. Following rehabilitation, the joints were able to bear weight again. After one to four years of follow-up, the patient satisfaction level was high, and the gait function of the patients improved a lot in all cases. Conclusions: Our results show that CT-based virtual surgical planning and, based on it, the use of a patient-specific 3D printed aiming device is a reliable method for major hip surgeries with significant bone loss. This technique has also made it possible to perform these operations with minimal X-ray exposure.

## 1. Introduction

In 1964, Ring [1] introduced the concept of a stemmed cup [1,2]. Ring’s method promised a novel way to increase the stability of the cup due to the concern that if a hemispherical cup was implanted into a hemispherical cavity, it could easily become loose if situated in the same geometrical environment; thus, the cup was stabilized with a stem.

Ring’s cup and its modifications have been tested by many researchers [2,3,4,5,6], but, ultimately, the stem of the original design became unnecessary in everyday practice, as there was no risk of loosening during subsequent practice. So, the usage of hemispherical cups became a standard treatment in hip endoprothetics.

However, the concept is not entirely outdated, and some manufacturers still maintain a model of this type in their catalogs [7].

In current practice, there is no need for such a model in primary routine arthroplasties. However, such periacetabular bone defects (Paprosky ≥ 2B) are becoming increasingly common, which is a significant degree of bone deficiency in the hip area by the most popular and widely used classification system.

In these revision cases, too many additional materials (auxiliary implants and large amounts of bone substitute) are needed to fill the gap between the cup and the host bone, and each of these increases the risk of complications.

So, the main clinical problem was to solve cases of severe periacetabular bone defects (Paprosky ≥ 2B) with reduced use of auxiliary implants and large amounts of bone substitute.

The revision of acetabular cups and tantalum augments offers a widespread option. For more important bone loss, custom-made [8] titanium implants are unlikely to become standard due to their long manufacturing time and high cost [9]. In addition, it is anticipated that the new European Union Medical Device Regulation 2017/745 will make the use of custom-made implants more difficult through the medical device certification procedure under the recently entered into-force regulation [10,11].

The surgical concept was that a single use of a stemmed cup, rather than using auxiliary implants, could solve the problem of hip recovery in these difficult situations. In this way, a single monobloc implant is sufficient, reducing the risk of implant-related complications.

This technique also allows the adjunct use of synthetic bone substitution materials, as a stemmed cup can keep any bone grafting material under pressure that may be used, thus achieving the concept of impaction grafting technique [12,13].

However, in cases of large bone defects, the instrumentation of the stemmed cup recommended for primary surgery cannot be used due to the frequent lack of standard bony reference points. Therefore, the use of a continuous X-ray image intensifier was necessary for the safe placement of the guide wire, which is essential for stemmed cup implantation.

So, the secondary clinical problem was to be able to safely place the guide wire without the use of a continuous image intensifier in order to reduce the radiation exposure suffered by the patient. A specific method has, therefore, been developed for these operations.

When the radiographs showed that the bone defect was more significant, and according to our experience, the situation seemed suitable for the use of a stemmed cup, the surgery was performed in two steps.

Accordingly, we utilized a method based on a thin slice CT scan of the pelvis to create a targeting device that allows the surgeon to safely insert a stable guidewire in the right position, made of 3D printable, sterilizable, biocompatible material.

The question was raised whether virtual surgical planning and the resulting patient-specific targeting instrument could be used to facilitate the performance of pelvic surgery in the severity of cases presented?

This publication reports on our initial experience and early results with stemmed cups for revision purposes [14,15,16].

## 2. Materials and Methods

Informed consent approved by the Ethical Board was obtained from all subjects involved in the study. This study has been developed in accordance with the relevant regulations.

If a larger-than-expected bone defect was found during revision surgery and it was determined that the use of a stemmed cup might be sufficient for reconstruction, the first step was the removal of the implants.

At the end of the operation, two cancellous self-tapping titanium screws (l = 30–40 mm, d = 6 mm) were percutaneously inserted into the iliac crest, one close to the anterior edge and another approximately 3 cm dorsally. Standard bone screws available in operating theatres were selected. The heads of the screws were left at approximately one-thread pitch above the bone surface, which ensures that the arms of the aiming device could firmly engage the bolt heads.

A thin-layer CT scan of the pelvis was performed a few days after the operation, allowing 3D planning of the correct position and size of the stemmed cup. Based on the two marker screws and one more periacetabular bony part, an aiming device was designed for the insertion of the guide wire. A sterilizable aiming device was 3D printed, and the cup was inserted after about 3 weeks.

A 3D pelvis reconstruction allowed further processing using the Mimics Innovation Suite (Materialize NV, Leuven, Belgium), which is an engineering software package that enables 3D medical imaging and surgical-related planning. So, the design of the patient-specific aiming device was based on a 3D reconstruction of the patient’s own anatomy. After importing and transforming the two-dimensional Digital Imaging and Communications in Medicine (DICOM) standard file, the radiation absorption range was selected using the windowing method for spatial imaging of the reliable bone structure (Figure 1).

The default threshold window ranges from 200 to 2000 HU according to the Hounsfield scale, modified to consider the parameterization of the recordings and the specific density of the selected bones.

The selected range is immediately displayed on the screen so that the consequence of the change in the threshold values can be immediately evaluated on the CT slice images in the form of a colored highlight.

The mask obtained in this way contained only the real bone volume relevant to the task, in our case, the pelvis with significant bone defects and the femur on the same side.

Despite the use of the artifact reduction algorithm, the two-dimensional mask generated from the raw images could contain harmful scatter or shadows. To reduce these, the reduce scatter function could be used as necessary [17].

This scatter reduction algorithm reduces the harmful effects of artifacts caused by metal objects in the field of view of CT images. By defining the source of the artifacts in a separate mask, it is possible to parameterize the filtering, the preview of which is immediately displayed in the program relative to the initial state.

After smoothing the mask to remove surface roughness due to the low resolution of the diagnostic imaging, the hemi-pelvis with significant bone defect was printed using a Stratasys F270 (Stratasys Ltd., Rehovot, Israel) fused deposition modeling (FDM) 3D printer. A square grip was attached at the sacrum in such a way that the model could be exposed on a stand and used as a reference to facilitate the spatial orientation in the OR.

The main goal was the safe placement of the guiding wire from a predetermined point in such a way that the cup’s stem should not penetrate the surface of the cortical bone towards the pelvic cavity (linea terminalis).

Once this guide wire is fixed in the correct position, the operation may be performed quasi-blindly, requiring only brief glances at the X-ray image intensifier.

### 2.1. Engineering Design

The purpose of the design was to create a targeting device that allows the surgeon to safely insert a stable guidewire in the right position, made of a 3D printable, biocompatible material.

The aiming devices were built in the 3-Matic (Materialize NV, Leuven, Belgium) medical-related CAD design software. After importing the hemipelvis, the most suitable cup diameter could be determined. This requires the editing of the largest sphere that can be modeled into the damaged side of the iliac crest.

These 3D models of stemmed cups of different sizes can be imported at scale for visualization and validation purposes. The design software allows the models to be displayed as semi-transparent, which visualizes any eventual overlaps. This can be particularly advantageous to ensure the highest level of bony contact with the cup. Special attention has been paid to the inclination of the stem.

The most appropriate stem length and position could then be determined without penetrating the cortical bone of the linea terminalis [18]. The goal was to have at least 2–3 mm of bone all around the stem of the cup (Figure 2a).

Next, the aiming device was designed, starting with the guiding sleeve. Its length was typically 60–70 mm, which provides appropriate guidance. However, this length is influenced by the position of the femur, which can be seen in the 3D reconstruction. Even with a diameter of 10–12 mm, the guide wire is safely targeted (Figure 2b). A chamfer of 1–2 mm around the inlet surface of the guide cylinder ensures that no cutting edge is formed at the boundaries of the surfaces. On the outlet side, also chamfered, was left 2–3 mm from the bone surface, thus avoiding the possibility of the guide stem accidentally resting on the surface of the shaft, thus altering the aiming direction. In the early versions, the targeting body adopted the unique shape of the damaged side of the hip, with a groove for the index finger to help hold it in position. In practice, however, it proved difficult to drill and hold the finger in a safe position at the same time. Furthermore, it was found to be problematic when the targeting device was in contact with the bone over a large surface area. The accuracy of the 3D reconstruction and the preparation of the hip may have had a negative effect on the direction of the targeting device in the event of large surface contact and, thus, the accuracy of aiming. These observations led to further development. Later practice was to position the target with three reliable bony sections. Once the support points were designated, slots were constructed between the designated points and the central part of the device. A hexagonal and a triangular socket was constructed to replace the finger positioning. A stainless-steel hand tool was used to retain it in position. However, the hand tool that was used limited the space available for the drilling tool, making it difficult to use the targeting device.

Our previous practice was to use three reliable bony formations to position the aiming device, but experience has shown that these can break off during surgery. However, in addition to two titanium screws, a third point on the lower bony frame provided a secure three-point fixation. For the ideal adjustment of its position, the support arms ending in 12 mm diameter spheres wrapped around the head of the two reference screws, and the third one touched a reliably solid part of the lower edge of the bony acetabulum. The edited spheres, using a Boolean subtraction operation, take the shape of the selected periacetabular bony anatomical point on the anterior margin of the defective acetabulum and the screw heads. Grooves were created in the spheres for the screw heads, considering the need to be able to remove the targeting device after drilling while keeping the guiding wire in position.

These three reference points were linked to each other and to the guide sleeve by means of suitably designed connecting profiles. It was considered that these reference points would be deep and difficult to reach during the operation, so the aiming cylinder of the guiding wire and the spheres around were connected by means of half-toroids of 50–60 mm diameter, considering bypassing the soft tissue (Figure 2c). In the last, fourth version, to replace the hexagonal and triangular sockets, a handle was added to the aiming device outside the skin surface (Figure 2d).

Therefore, the aiming device was first validated on a virtual 3D model. The entry and exit points and the path it takes through the bone were clearly visible in the design software, using the semi-transparent display mode. Then, the 3D-printed targeting model was fitted to the previously printed hemipelvis with its actual dimensions. The accuracy of the aiming was also verified by test drilling. The assembled model was then available to the surgeon in the OR for visual inspection, which helped to ensure spatial orientation (Figure 3).

After approval, a label with a unique identifier was created on the 3D model, and then the aiming device was printed using a sterilizable, biocompatible, rigid, transparent photopolymer material (MED-610, Stratasys Ltd., Rehovot, Israel) [19].

Following 3D printing, the support material was removed by high-pressure waterjet and brush cleaning, followed by post-processing of the workpieces according to the manufacturer’s instructions.

### 2.2. Surgical Technique

In accordance with the practices of our department, a so-called modified Watson–Jones approach was used, which provides adequate access to the target area from the anterolateral side for this type of surgical procedure.

Once the bony acetabulum was cleared, soft tissues were removed around the anchor points of the aiming device, and it was fitted on the two screw heads and the third bony reference point.

The guiding wire was inserted and controlled with fluoroscopy flashes (Figure 4). The drilling and reaming were carried out in accordance with the recommendations:

The direction of the cup stem was determined by the guide wire. The aiming device was needed to introduce this wire. After removing the aiming device, the next step was the drilling. Since a cannulated drill bit was used, the previously installed wire could guide the process. A self-positioning reaming tool was then used in the drilled channel. In the prepared cavity, the cup was fitted perfectly. Of the two stemmed cups available to us, the McMinn cup (Waldemar Link, Hamburg, Germany) had the simpler geometry and was therefore chosen. The stemmed cup was inserted according to the manufacturer’s recommendation. In the presence of a significant bone defect, a synthetic bone graft may be impacted for substitution.

The criterion of a successful revision is to make the hip joint load-bearing again. Using our technique, with proper design, sizing, and insertion, sufficient primary stability can be achieved.

Since these were very serious cases with very specific problems, postoperative rehabilitations were always determined individually by the rehabilitation specialists.

In general, only unloaded mobilization was allowed for 10 days, and then the load was increased to 5 kg increments per week until full load was reached. Walking is initially allowed with a walking frame or two crutches.

Between February 2018 and July 2021, revision surgeries of large periacetabular defects (Paprosky ≥ 2B) were performed with a stemmed cup using the above-described method in a total of 17 patients. The age of the patients ranged from 35 to 77 years, the gender distribution was 12 females and 5 males, and the follow-up period ranged from 10 months to 34 months. All patients belonged to the Caucasian race. See Table 1 for complete details.

The patients did not have any other conditions apart from musculoskeletal problems that could have compromised the outcome of the surgeries. All patients had major bone defects that could not have been resolved by conventional methods using a single implant.

The radiological examination allowed us to verify the close bone-to-implant contact and the unchanged position of the implant during follow-up.

For primary hip replacements, the Harris Hip Score (HHS) is used to assess hip function before and after surgery. In these severe revision cases, it is not relevant. It can be explained as follows: This group of patients had been severely limping for a significant time prior to surgery. These patients are instructed to expect partial weight bearing after surgery. Therefore, they are fundamentally different from patients in whom primary hip arthroplasty is routinely performed for osteoarthritis.

## 3. Results

In all the cases operated with the above-described targeting procedure, the stems of the cups remained between the cortical bone surfaces without perforation of the linea terminalis, as shown by postoperative radiographs. There were no complicated surgical situations. In 16 cases, the wound healings were uneventful, and the hips were able to bear weight again after postoperative rehabilitation.

The only patient who suffered an infection healed after the removal of the implant. This complication rate is known and in line with accepted values.

One septic complication occurred. This complication rate is consistent with other revision techniques. In this case, the implant had to be removed, and the patient remained in Girdlestone condition. In such cases, removal of the implants and, of course, surgical debridement and antibiotic treatment are essential. The condition after resection is called Girdlestone’s condition. We had no other complications or unexpected outcomes.

At the time of the manuscript edition, some cases reported in this study were still in the early postoperative period. Even the shortest 10-month follow-up means that patients’ wound healing was uninterrupted, and the implant was securely fixed. By this time, patients are beyond successful rehabilitation. Of course, as with all such patients, monitoring will continue at annual check-ups. Report on the mid and long-term outcomes, gait analysis, and subjective assessments of patients is planned.

However, since our surgical experience was positive in terms of early postoperative functional and radiological results, it was considered worthwhile to demonstrate the technical support for the developed method, which is accessible to anyone and provides effective, safe surgical outcomes.

## 4. Discussion

Hip arthroplasty has entered a new era in the last decade. Life expectancy has increased, and along with this, the wish for a better quality of life arose [20]. The inevitable prosthesis wear, loosening, and various complications, as well as possible accidents experienced by prosthesis wearers, have directed the profession away from surgical abstention. Nowadays, we are undertaking more and more complicated revisions.

The solution always depends on the case itself: the patient’s general condition, needs, expectations, and level of cooperation, as well as on the surgeon’s abilities, experience, and knowledge. Therefore, there are different ways of solution. In severe cases, trabecular metal augments or antiprotrusion cages are also used, but no excellent surgical solution exists [21,22,23,24].

Most currently used methods basically use implants that are fixed to the bone with screws. Due to the shear forces that may occur, screw fixation may cause implant failure in the long term. The stemmed cup is fixed with a press fit within the bone.

Two-step surgery was used for greater safety. On the one hand, to avoid possible septic complications, and on the other hand, during the first operation, we have a more accurate view of the amount of bone that can be used to fix the implant.

One more argument for the case of two-step acetabular revisions with large bone defects is that the bone structures depicted on the imaging devices may appear accurate based on their radiopaque properties but are often no longer suitable for fastening screws with sufficient force, which is mandatory to success. However, during the first operation, the bone quality of the acetabular environment can be accurately assessed from a mechanical point of view.

However, most offer acetabulum reconstruction based on plate-and-screw fixation analogous to methods used in trauma surgery.

The development of our method was focused on the revision using a stemmed cup fixed in the ilium.

As this method was undergoing development, practically every detail of the aiming device was modified case by case for the first few operations, as presented in this study. As there were no cases of misdirection, our current method has proven to be safe, reliable, and simple to apply.

In this study, the planning and realization of our surgical technique were presented, which has been refined since the initial surgeries and has become routinely consistent.

However, possible future directions of targeting are already visible: research opportunity to develop a system that uses augmented reality (AR) technology to significantly enhance targeting during surgeries requiring high accuracy [25]. AR systems would allow the surgeon to see a real-time image of the surgery through the glasses, without the need for complex navigation techniques, where a virtual model of the bone reconstructed from the CT image and the instrument used (drill, screw, screwdriver) would be displayed in the real surgical environment [26,27,28].

Despite its history of only a few years, augmented reality is already being used in several surgical applications, with a variety of experimental solutions for the overlaying of 3D virtual space and real anatomical structures, the so-called registration, but the accuracy still typically varies between 3 and 6 mm, which is not sufficient for the higher precision procedures in question [29]. The procedure described above requires the registration process to be performed using a high-precision 3D scanner with sub-millimeter accuracy [30,31].

This allows the virtual and real bone to be aligned much more accurately than existing methods and eliminates the need for time-consuming manual registration, which saves additional time [32].

## 5. Conclusions

The dual objective was successfully reached. On the one hand, it was possible to operate on these severe cases with a single implant (stemmed cup) instead of multiple implants, and on the other hand, we have resolved the main technical difficulty of the stemmed cup insertion without the continuous use of an X-ray image intensifier with the targeting device for such severe cases.

Our results so far show that CT-based virtual surgical planning and, based on it, the usage of a patient-specific 3D printed targeting device is a reliable method for pelvic surgery. All this is achieved with an order of magnitude lower radiation exposure than before.

Our experiences give promise for the application of this method to the planning and safe performing of complex, multiple pelvic fracture surgeries. This offers further potential, paving the way for the design, fabrication, and safe placement of custom-made implants.

The relatively small number of cases and the highly variable follow-up time do not yet allow long-term conclusions to be drawn. However, it is believed that others may reproduce the technique we can achieve with great precision without relying on sophisticated navigation methods.

A clear benefit of the procedure is that there is no need to make ad hoc decisions when positioning the implant. The targeting device ensures that the implant is placed in the planned position, and we know the size of the implant to be chosen in advance, thanks to the CT-based 3D design option.

By using this patient-specific targeting device, complications such as vascular, nerve, or intestinal injuries resulting from misalignment could be safely avoided.

While further modifications may be made to our approach, our current solution is offered as a reference for consideration by those in the field.

No complications were found in any patient since the manuscript was closed. A safe way of inserting stem cups in cases with significant bone loss has been solved, eliminating the risk of pelvic perforation.

## Figures and Tables

**Figure 1 bioengineering-10-01095-f001:**
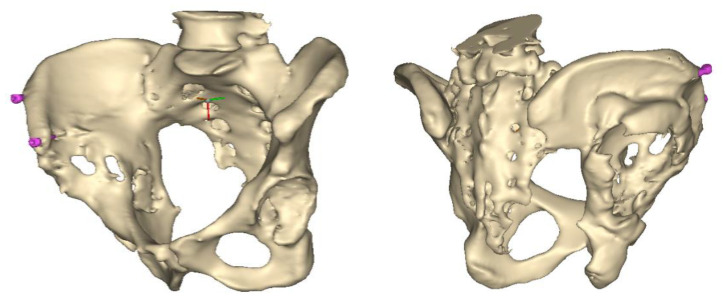
3D images of the mask that appeared when the CT scans were threshold and the spatial model derived from it. The heads of the implanted screws are displayed in purple.

**Figure 2 bioengineering-10-01095-f002:**
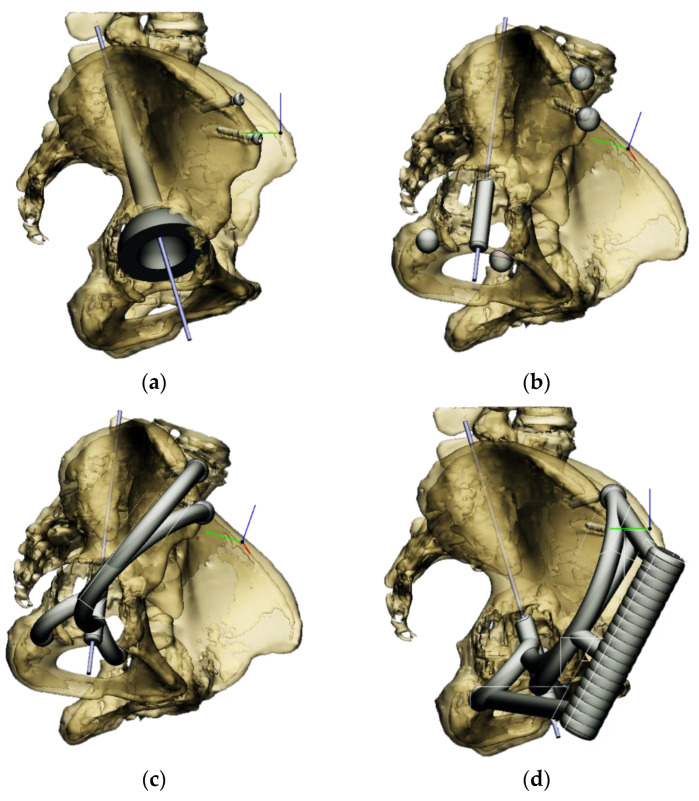
Design steps of the aiming device: (**a**) Stemmed acetabular cup with guiding wire; (**b**) drawing of the guiding sleeve and the reference points; (**c**) connection of the reference points and the guiding sleeve; and (**d**) adjustment of the handle.

**Figure 3 bioengineering-10-01095-f003:**
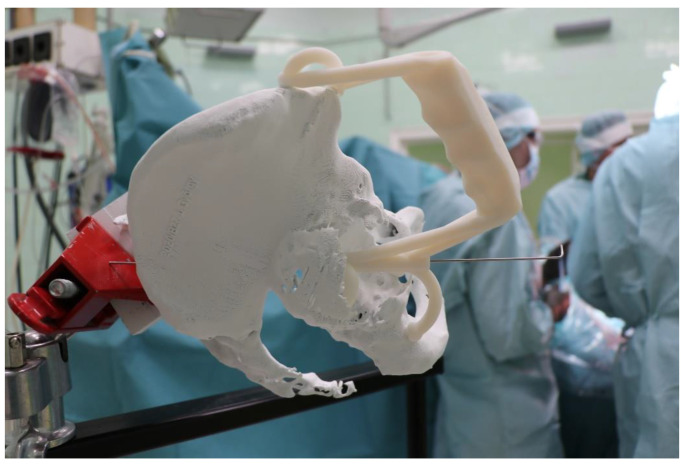
The 3D-printed hemipelvis with the aiming device in the OR.

**Figure 4 bioengineering-10-01095-f004:**
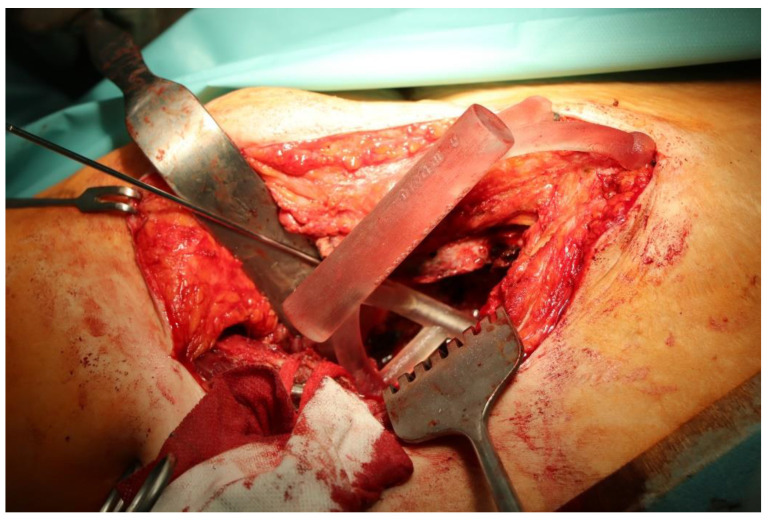
The biocompatible aiming device after insertion of the guiding wire.

**Table 1 bioengineering-10-01095-t001:** Performed revision surgery cases of large periacetabular defects with stemmed cups using our method.

No.	Age/Sex	Follow-Up Time/Classification	Preop X-ray	Interim X-ray	Interim X-ray	Last X-ray	3D Printed Hemi-Pelvis	The Aiming Device
1	35 years♀	47.5 months 3B	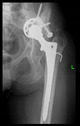	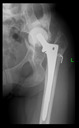	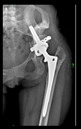	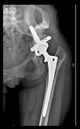	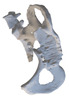	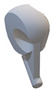
2	63 years ♂	45.5 months 3B	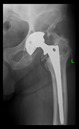	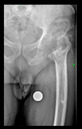	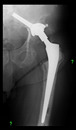	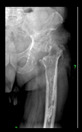	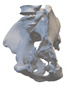	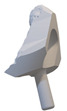
3	54 years♀	39.5 months 3A	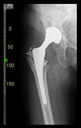	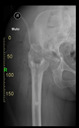	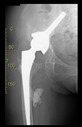	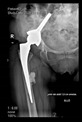	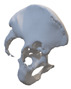	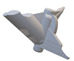
4	72 years♀	38 months 2C	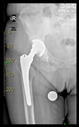	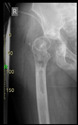	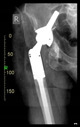	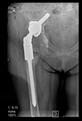	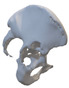	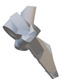
5	65 years♀	34 months 3B	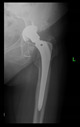	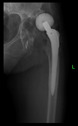	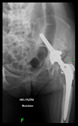	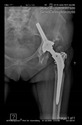	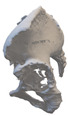	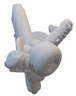
6	62 years♀	31 months 3A	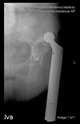	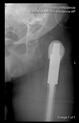	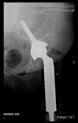	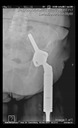	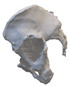	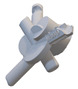
7	77 years ♂	27.5 months 3B	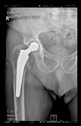	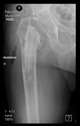	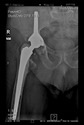	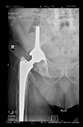	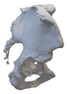	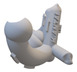
8	66 years♀	24.5 months 3A	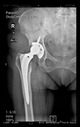	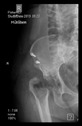	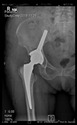	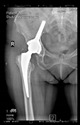	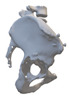	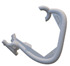
9	60 years♂	23 months 3A	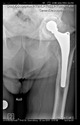	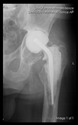	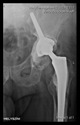	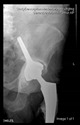	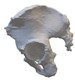	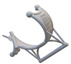
10	67 years♀	18 months 3A	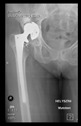	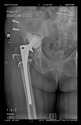	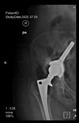	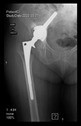	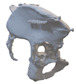	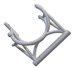
11	63 years♀	16.5 months 2B	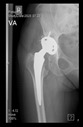	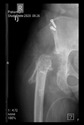	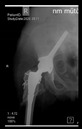	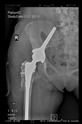	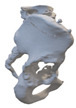	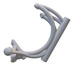
12	66 years♀	16 months 3A	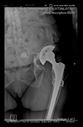	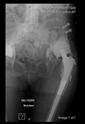	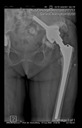	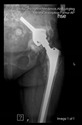	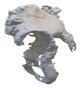	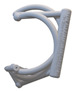
13	53 years♀	15 months 3A	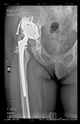	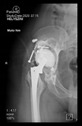	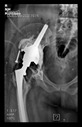	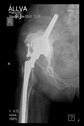	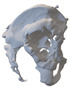	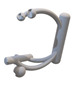
14	57 years♀	12 months 3A	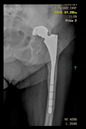	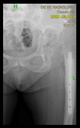	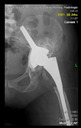		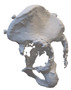	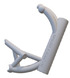
15	65 years♂	11 months 2B	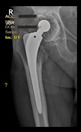	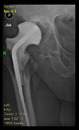	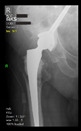		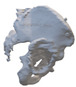	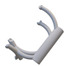
16	66 years♀	10 months 3A	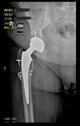	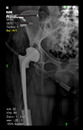	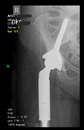		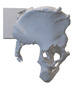	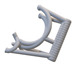
17	71 years♂	10 months 2B	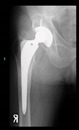	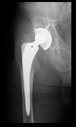	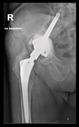		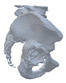	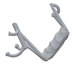

## Data Availability

The data presented in this study are available on request from the corresponding author.

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
