# Peer review of "Acetabular Revision with McMinn Cup: Development and Application of a Patient-Specific Targeting Device"

_bioengineering, 2023, doi:10.3390/bioengineering10091095_

Round 1

Reviewer 1 Report

The title seems to provide an accurate description of the content but could be considered technical for readers not familiar with the field.

Engagement: Depending on your target audience, you might consider a title that better highlights the unique contribution or innovation of the research. This could help draw a broader audience or emphasize the importance of your work.

A possible revision could be:

"Acetabular Revision with McMinn Cup: Development and Application of a Patient-Specific Targeting Device"

For Abstract

1. Content and Clarity:

Expand on the surgical complication mentioned ("septic reason") to clarify its nature and relevance. Details may help readers understand the full scope of the outcomes.

You may want to make explicit any quantitative measures of success (e.g., a statistical analysis of improved gait function) to strengthen the conclusion.

2. Structure and Flow:

The abstract is quite dense and may benefit from more structured formatting. Consider dividing it into separate sections, or providing more line breaks to visually separate different parts of the research (Background, Methods, Results, Conclusions).

For Introduction

1. Content and Clarity:

The historical context provided is good, but it may be enhanced by explaining why the stemmed cup method lost its prominence in everyday practice.

Clarify the specific challenges associated with large bone defects and why existing methods are not sufficient.

Elaborate on what makes the McMinn cup a viable solution, focusing on its features or properties that allow it to address the mentioned challenges.

The use of synthetic bone substitution materials is mentioned but not explained. Additional context or explanation would provide better understanding.

Explain any technical terms or jargon, such as the significance of "Paprosky ≥ 2b," for readers who may be less familiar with these terms.

The information about the European Union Medical Device Regulation 2017/745 could be expanded to explain how it affects the use of custom-made implants.

2. Structure and Flow:

Consider restructuring the introduction to move from a historical overview to a discussion of current challenges, followed by a description of the authors' approach.

The transition between discussing the McMinn cup and the introduction of new regulations could be more seamless.

Some paragraphs could be further broken down into more concise sections to guide the reader through the content more comfortably.

For Materials and Methods

1. Content and Clarity:

The surgical process is described in detail, but certain terms and methods might require clarification for readers who are not experts in the field.

Consider explaining why the specific screw dimensions were chosen.

Elaborate on the significance of leaving the screw heads "one-thread pitch above the bone surface."

Explain the rationale behind each step, such as why two screws were inserted, the importance of 3D pelvis reconstruction, or the specific choice of engineering software.

Consider adding information about the criteria for selecting the McMinn cup for reconstruction.

Define or describe specialized terms like "linea terminalis."

2. Structure and Flow:

Consider reorganizing subsections or introducing additional headings to guide the reader through the process.

The transition from the general method to the engineering design and surgical technique should be smooth, with clear connections between the different stages.

3. 2.1. Engineering Design:

Clarify the design goals and constraints, including why certain dimensions or materials were chosen.

Detail the validation process for the engineering design, explaining how the aiming device's functionality was ensured.

Discuss any challenges or innovations in the design process, such as unique solutions to fitting the guide wire or ensuring bony contact with the cup.

4. 2.2. Surgical Technique:

Provide more details about the Watson–Jones approach, explaining why it was chosen and what it involves.

Elaborate on the steps for inserting the guide wire, drilling, reaming, and cup insertion, explaining the rationale behind each stage.

Explain the criteria for successful surgery and how postoperative rehabilitation is managed.

If any postoperative complications or challenges can arise, these might be worth mentioning.

5. Ethical Considerations:

If human subjects were involved, consider adding a statement about ethical considerations, including informed consent, compliance with relevant regulations, or oversight by an institutional review board.

For Results

1. Content and Clarity:

Patient Information and Details: Include details about the patients, such as age, gender, underlying conditions, etc. This will provide context for the type and extent of the defects being treated.

Implant Removal and Complications: Elaborate on the circumstances leading to the removal of the implants and the Girdlestone condition. Clarify what measures were taken to address the septic complication, and if there were any other complications or unexpected outcomes.

Comparison with Other Methods: If possible, include comparisons to other methods or techniques used for similar conditions, with appropriate references. This could provide valuable context to the efficacy and advantages of the described method.

Harris Hip Score (HHS): Explain why the HHS is not relevant to this particular group of patients, and if any other scoring or evaluation methods were used to assess patient outcomes.

Long-term Follow-up: Clearly state the time frame for follow-up and define what is meant by "early postoperative period." Provide an anticipated timeline for reporting on mid- & long-term outcomes.

Technical Support and Outcome Measures: Provide more specific details about the technical support provided by engineers, and clearly define what was considered as favorable in terms of early postoperative functional and radiological results.

2. Structure and Flow:

Consider breaking down the section into subheadings, such as "Patient Demographics," "Surgical Outcomes," "Complications," "Functional and Radiological Results," etc., to guide the reader through the various aspects of the results.

Ensure smooth transitions between paragraphs to maintain a cohesive narrative.

3. Relevance to Objectives:

Reflect back on the objectives or research questions posed in the introduction or methods sections, and directly address how the results provide answers or insights into those areas.

4. Ethical Considerations:

Reiterate or refer to any ethical considerations related to patient data, privacy, and informed consent.

5. Conclusions within Results:

The last paragraph seems to draw conclusions from the results. While this is not necessarily a problem, you may want to consider whether these statements would be better placed in a separate "Discussion" or "Conclusion" section.

For Discussion

1. Introduction to the Discussion:

The opening lines set the stage for the challenges faced in hip arthroplasty, acknowledging the changes over the last decade. Consider adding some references to the literature that support these statements.

2. Case-by-Case Consideration and Tailoring:

The section describes the multifaceted considerations needed for each patient and how solutions vary. More elaboration and examples may enrich this point, showcasing the complexity of the process.

3. Argument for Two-Step Acetabular Revisions:

This paragraph introduces an interesting concept but needs further development. What are the benefits or disadvantages of two-step revisions? How does this apply to your method specifically?

4. Comparison with Existing Methods:

The comparison to web-based advertisements and trauma surgery techniques is a bit vague. It would be more informative to compare your method to other established surgical techniques, describing the pros and cons and where your method may fit in or excel.

5. Details of Your Method and Aiming Device:

You've outlined the evolution of your method and aiming device. Consider including a sub-section or separate paragraph that explains what specific challenges were addressed with each modification. Visual aids or diagrams might be particularly helpful here.

6. Safety, Reliability, and Simplicity:

These claims could be strengthened with more detailed evidence from your study results. How did you measure these attributes, and what were the specific findings that led to these conclusions?

7. Limitations and Future Directions:

Discuss any limitations of the method or the study itself, such as small sample size, short follow-up period, etc.

Indicate the potential for future research, improvements, or applications of the method.

8. Connection to Other Sections of the Paper:

Make sure to tie the discussion back to the specific results and methodology you've detailed earlier in the paper, showing how they all support the conclusions you're drawing here.

9. Concluding Remarks:

Consider ending with a strong concluding statement or paragraph that summarizes the main contributions and implications of your work, and how it may influence future clinical practice or research.

For Conclusions

1. Acknowledgment of Limitations:

The section begins with an acknowledgment of the limitations related to the small number of cases and variable follow-up time. This is a strong point and adds credibility to the paper. It may be helpful to expand slightly on how these limitations affect the conclusions, or what specific insights might be gained with more data or a longer follow-up.

2. Main Conclusion:

The main conclusion is that the approach described in the paper can be reproduced without relying on sophisticated navigation methods. This is a clear statement, but it could be enhanced by summarizing the key benefits or features of the method. What makes it particularly valuable, effective, or innovative?

3. Future Modifications and Current Solution:

This statement provides an open door for future improvements while also standing by the current solution. It might be beneficial to discuss what specific areas may be open to further modification and why the current solution is valuable as a reference.

4. Connection to Broader Context:

While the conclusion is concise, it could be strengthened by tying it back to the broader context and goals of the work. What problem was being solved, and how does this solution contribute to the field?

5. Recommendations or Implications:

Consider discussing the potential implications or applications of this work. How could it influence clinical practice, or what further research might be conducted based on these findings?

For abstract:

There are some minor grammatical and wording issues throughout the abstract that could be revised for better clarity. For example, "the implantation for cases where" could be revised to "the implantation in cases where."

Consider revising "what is the clue of the implantation" to something more clear, such as "which is essential to the implantation process."

For Introduction

Watch for typos like "loose" instead of "lose."

Use consistent tense and voice throughout.

The flow between sentences can be smoothed to provide a clearer progression of ideas.

For Materials and Methods

Be consistent with tense (e.g., "a thin layer CT scan of the pelvis is performed" vs. "was performed").

Check for missing or unnecessary commas.

Address any spelling or typographical errors.

Ensure that the language is precise and clear, avoiding ambiguous terms or phrases.

Author Response

Dear Reviewer!

Thank you for your thorough review!

We have tried to improve the quality of our manuscript according to your suggestions and taking them into account.

We would like to note that unfortunately we realized too late that the Bioengineering readership is partly or mostly composed of technical experts. Consequently, we did not take this into account in the previous version.

Yours sincerely,
The authors

The title seems to provide an accurate description of the content but could be considered technical for readers not familiar with the field.

Engagement: Depending on your target audience, you might consider a title that better highlights the unique contribution or innovation of the research. This could help draw a broader audience or emphasize the importance of your work.

A possible revision could be:

"Acetabular Revision with McMinn Cup: Development and Application of a Patient-Specific Targeting Device"

We have changed the title.

For Abstract

  1. Content and Clarity:

Expand on the surgical complication mentioned ("septic reason") to clarify its nature and relevance. Details may help readers understand the full scope of the outcomes.

We have extended the sentence to clarify this septic situation is outstanding.

You may want to make explicit any quantitative measures of success (e.g., a statistical analysis of improved gait function) to strengthen the conclusion.

The Harris Hip Score could be a quantitative measure, but it is only meaningful for routine, primary osteoarthritis cases. It was described in the Results section.

  1. Structure and Flow:

The abstract is quite dense and may benefit from more structured formatting. Consider dividing it into separate sections, or providing more line breaks to visually separate different parts of the research (Background, Methods, Results, Conclusions).

We have used the four sections you mentioned to separate the abstract.

For Introduction

  1. Content and Clarity:

The historical context provided is good, but it may be enhanced by explaining why the stemmed cup method lost its prominence in everyday practice.

We have added a tagline to explain why the stemmed cup lost its importance.

Clarify the specific challenges associated with large bone defects and why existing methods are not sufficient.

We have added a sentence to clarify the situation.

Elaborate on what makes the McMinn cup a viable solution, focusing on its features or properties that allow it to address the mentioned challenges.

We have added a sentence next to the previously added one to explain the importance of our choose.

The use of synthetic bone substitution materials is mentioned but not explained. Additional context or explanation would provide better understanding.

We have extended the sentence to explain this one.

Explain any technical terms or jargon, such as the significance of "Paprosky ≥ 2b," for readers who may be less familiar with these terms.

We have added a tagline after Paprosky 2b to explain this.

The information about the European Union Medical Device Regulation 2017/745 could be expanded to explain how it affects the use of custom-made implants.

We have added a tagline to clarify challenges with the MDR what caused that Europe is facing a medical device gridlock as medical companies wait to have all their products certified.

  1. Structure and Flow:

Consider restructuring the introduction to move from a historical overview to a discussion of current challenges, followed by a description of the authors' approach.

We believe that we have written it in line with your suggestions.

The transition between discussing the McMinn cup and the introduction of new regulations could be more seamless.

There are more options for dealing with cases of significant bone loss than the McMinn cup. A list of the disadvantages of these solutions leads to the MDR regulation.

Some paragraphs could be further broken down into more concise sections to guide the reader through the content more comfortably.

We have made it.

For Materials and Methods

  1. Content and Clarity:

The surgical process is described in detail, but certain terms and methods might require clarification for readers who are not experts in the field.

We have just increased understandability by using terms that are more widely understood.

Consider explaining why the specific screw dimensions were chosen.

We have added a sentence after the previously added one to explain our choose.

Elaborate on the significance of leaving the screw heads "one-thread pitch above the bone surface."

We have extended the sentence.

Explain the rationale behind each step, such as why two screws were inserted, the importance of 3D pelvis reconstruction, or the specific choice of engineering software.

We have added taglines and sentences to clarify the mentioned points.

Consider adding information about the criteria for selecting the McMinn cup for reconstruction.

A sentence were added to explain our choice.

Define or describe specialized terms like "linea terminalis."

We added a tagline to help the understanding.

  1. Structure and Flow:

Consider reorganizing subsections or introducing additional headings to guide the reader through the process.

We made it clearer for a better overview.

The transition from the general method to the engineering design and surgical technique should be smooth, with clear connections between the different stages.

The three phases have been described as the process has gone from case to case. There was no particular overlap between them, After the surgical planning and the decision of tecnique the engineering design and production was started. The next step was the operation.

  1. 2.1. Engineering Design:

Clarify the design goals and constraints, including why certain dimensions or materials were chosen.

We have extended a sentence to clarify our goals.

Detail the validation process for the engineering design, explaining how the aiming device's functionality was ensured.

We have extended the sentence about the validation process and added one more sentence to make it clearer.

Discuss any challenges or innovations in the design process, such as unique solutions to fitting the guide wire or ensuring bony contact with the cup.

We have added a paragraph to explain the challenge of the design process.

  1. 2.2. Surgical Technique:

Provide more details about the Watson–Jones approach, explaining why it was chosen and what it involves.

We have extended the sentence to explain the reason.

Elaborate on the steps for inserting the guide wire, drilling, reaming, and cup insertion, explaining the rationale behind each stage.

Sentences were added to explain steps of surgery.

Explain the criteria for successful surgery and how postoperative rehabilitation is managed.

We have added a sentence to state the criteria. One more sentence was added to describe how postoperative rehabilitation is managed in general.

If any postoperative complications or challenges can arise, these might be worth mentioning.

We have added a sentence to make it clear.

  1. Ethical Considerations:

If human subjects were involved, consider adding a statement about ethical considerations, including informed consent, compliance with relevant regulations, or oversight by an institutional review board.

We have added sentences to the beginning of the section about the informed consent and the compliance.

For Results

  1. Content and Clarity:

Patient Information and Details: Include details about the patients, such as age, gender, underlying conditions, etc. This will provide context for the type and extent of the defects being treated.

We have added more details to the first paragraph.

Implant Removal and Complications: Elaborate on the circumstances leading to the removal of the implants and the Girdlestone condition. Clarify what measures were taken to address the septic complication, and if there were any other complications or unexpected outcomes.

Sentences were added to explain this condition and the circumstances.

Comparison with Other Methods: If possible, include comparisons to other methods or techniques used for similar conditions, with appropriate references. This could provide valuable context to the efficacy and advantages of the described method.

A sentence were added with references to present other methods for severe cases at the discussion section.

Harris Hip Score (HHS): Explain why the HHS is not relevant to this particular group of patients, and if any other scoring or evaluation methods were used to assess patient outcomes.

The paragraph of HHS extended to explain the reason.

Long-term Follow-up: Clearly state the time frame for follow-up and define what is meant by "early postoperative period." Provide an anticipated timeline for reporting on mid- & long-term outcomes.

We have added a sentence to the paper to clarify the follow-up question.

Technical Support and Outcome Measures: Provide more specific details about the technical support provided by engineers, and clearly define what was considered as favorable in terms of early postoperative functional and radiological results.

We have added sentences to explain these results.

  1. Structure and Flow:

Consider breaking down the section into subheadings, such as "Patient Demographics," "Surgical Outcomes," "Complications," "Functional and Radiological Results," etc., to guide the reader through the various aspects of the results.

Sentences were added and reorganized to help the reader.

Ensure smooth transitions between paragraphs to maintain a cohesive narrative.

We have tried to make it even smoother.

  1. Relevance to Objectives:

Reflect back on the objectives or research questions posed in the introduction or methods sections, and directly address how the results provide answers or insights into those areas.

We added a paragraph to the end of the heading.

  1. Ethical Considerations:

Reiterate or refer to any ethical considerations related to patient data, privacy, and informed consent.

We’ve added a sentence to the beginning of the heading.

  1. Conclusions within Results:

The last paragraph seems to draw conclusions from the results. While this is not necessarily a problem, you may want to consider whether these statements would be better placed in a separate "Discussion" or "Conclusion" section.

Thank you!

For Discussion

  1. Introduction to the Discussion:

The opening lines set the stage for the challenges faced in hip arthroplasty, acknowledging the changes over the last decade. Consider adding some references to the literature that support these statements.

A reference were added to support the statements.

  1. Case-by-Case Consideration and Tailoring:

The section describes the multifaceted considerations needed for each patient and how solutions vary. More elaboration and examples may enrich this point, showcasing the complexity of the process.

This section was reworked and enriched with some additional informations to help understanding the challenges and the complexity of these cases.

  1. Argument for Two-Step Acetabular Revisions:

This paragraph introduces an interesting concept but needs further development. What are the benefits or disadvantages of two-step revisions? How does this apply to your method specifically?

We have added a sentence to explain benefits.

  1. Comparison with Existing Methods:

The comparison to web-based advertisements and trauma surgery techniques is a bit vague. It would be more informative to compare your method to other established surgical techniques, describing the pros and cons and where your method may fit in or excel.

We have removed that sentence you mentioned. For each of these serious cases, all the options need to be considered for the successful outcome. Reworked the section and mentioned common tecniques with references.

  1. Details of Your Method and Aiming Device:

You've outlined the evolution of your method and aiming device. Consider including a sub-section or separate paragraph that explains what specific challenges were addressed with each modification. Visual aids or diagrams might be particularly helpful here.

Table 1 provides a visual representation of the evolution of our targeting device. In materials and methods there is a paraghraph about the challenges.

  1. Safety, Reliability, and Simplicity:

These claims could be strengthened with more detailed evidence from your study results. How did you measure these attributes, and what were the specific findings that led to these conclusions?

We have added taglines to that sentence to explain.

  1. Limitations and Future Directions:

Discuss any limitations of the method or the study itself, such as small sample size, short follow-up period, etc.

In the conclusion section we have mentioned these limitations.

Indicate the potential for future research, improvements, or applications of the method.

A sentence was added about the future potentials (AR-VR).

  1. Connection to Other Sections of the Paper:

Make sure to tie the discussion back to the specific results and methodology you've detailed earlier in the paper, showing how they all support the conclusions you're drawing here.

We added a tagline to tie back.

  1. Concluding Remarks:

Consider ending with a strong concluding statement or paragraph that summarizes the main contributions and implications of your work, and how it may influence future clinical practice or research.

Section was reworked and now it has straight concluding sentences at the end.

For Conclusions

  1. Acknowledgment of Limitations:

The section begins with an acknowledgment of the limitations related to the small number of cases and variable follow-up time. This is a strong point and adds credibility to the paper. It may be helpful to expand slightly on how these limitations affect the conclusions, or what specific insights might be gained with more data or a longer follow-up.

The main conclusions remained straightforward. Later, with more data and more detail available through the gait-analysis and the annual monitoring, the long-term outcomes can be described.

  1. Main Conclusion:

The main conclusion is that the approach described in the paper can be reproduced without relying on sophisticated navigation methods. This is a clear statement, but it could be enhanced by summarizing the key benefits or features of the method. What makes it particularly valuable, effective, or innovative?

We have added a sentence to summarize the benefits.

  1. Future Modifications and Current Solution:

This statement provides an open door for future improvements while also standing by the current solution. It might be beneficial to discuss what specific areas may be open to further modification and why the current solution is valuable as a reference.

At the moment, we only see the need for small changes, like designing a more ergonomic handle that doesn't limit the working space of the drill, etc., so more design things.

  1. Connection to Broader Context:

While the conclusion is concise, it could be strengthened by tying it back to the broader context and goals of the work. What problem was being solved, and how does this solution contribute to the field?

We have reworked this section too. It has more strength now but also straight and honest enough.

  1. Recommendations or Implications:

Consider discussing the potential implications or applications of this work. How could it influence clinical practice, or what further research might be conducted based on these findings?

Over time, all surgical techniques are slowly becoming less sophisticated and simplified, and we expect the same here.

Comments on the Quality of English Language

For abstract:

There are some minor grammatical and wording issues throughout the abstract that could be revised for better clarity. For example, "the implantation for cases where" could be revised to "the implantation in cases where."

We have corrected those.

Consider revising "what is the clue of the implantation" to something more clear, such as "which is essential to the implantation process."

 We have corrected it.

For Introduction

Watch for typos like "loose" instead of "lose."

Use consistent tense and voice throughout.

The flow between sentences can be smoothed to provide a clearer progression of ideas.

 Corrections were applied.

For Materials and Methods

Be consistent with tense (e.g., "a thin layer CT scan of the pelvis is performed" vs. "was performed").

Check for missing or unnecessary commas.

Address any spelling or typographical errors.

Ensure that the language is precise and clear, avoiding ambiguous terms or phrases.

Correction were applied too.

Reviewer 2 Report

The manuscript describes an approach to solve an important issue. Hip arthroplasty has entered a new era in the last decade. Due to the increase in life expectancy there is the need of better quality of life. The inevitable prosthesis wear, loosening, and various complications as well as possible accidents experienced by prosthesis wearers have directed the profession away from surgical abstention. Nowadays clinicians have to face more and more complicated revisions. Therefore there is the need to overcome these problems.

Although the study presents a great variability, it is an important work that can serve as a basis for other studies, as well as procedures in similar situations.

Some minor changes should be done in order to publish the manuscript:

1)      Page 9 line 134: The sentence “All operations were performed by the first author” is not necessary here. The authors should mention that in the “Author Contributions” section

2)      Throughout the entire document authors have used first person plural. This should be changed. For example in the abstract (page 1, line 9) instead of “Next, by applying the metal artifact removal (MAR) algorithm, we performed a CT scan of the pelvis”  it should be “Next, by applying the metal artifact removal (MAR) algorithm, a CT scan of the pelvis was performed”. This should be done for the entire document.

After these changes the work is publishable.

Author Response

Dear Reviewer!

Thank you for your thorough review!

We have tried to improve the quality of our manuscript according to your suggestions and taking them into account.

Yours sincerely,
The authors

The manuscript describes an approach to solve an important issue. Hip arthroplasty has entered a new era in the last decade. Due to the increase in life expectancy there is the need of better quality of life. The inevitable prosthesis wear, loosening, and various complications as well as possible accidents experienced by prosthesis wearers have directed the profession away from surgical abstention. Nowadays clinicians have to face more and more complicated revisions. Therefore there is the need to overcome these problems.

Although the study presents a great variability, it is an important work that can serve as a basis for other studies, as well as procedures in similar situations.

Thank you!

Some minor changes should be done in order to publish the manuscript:

  • Page 9 line 134: The sentence “All operations were performed by the first author” is not necessary here. The authors should mention that in the “Author Contributions” section

We have changed it.

  • Throughout the entire document authors have used first person plural. This should be changed. For example in the abstract (page 1, line 9) instead of “Next, by applying the metal artifact removal (MAR) algorithm, we performed a CT scan of the pelvis”  it should be “Next, by applying the metal artifact removal (MAR) algorithm, a CT scan of the pelvis was performed”. This should be done for the entire document.

We have corrected those.

After these changes the work is publishable.

Thank you!

Reviewer 3 Report

The following points should be addressed before the paper can be considered for publication

Abstract

·         What is the clinical problem? What is the concept of the surgical solution? What is the hypothesis? What are the results? – the abstract should be completely rewritten

Introduction

·         “so-called” – change to “a concept”

·         “Developed during the  golden age of hip endoprosthetics,” – remove

·         “The idea was based on” – change to “due to” and rephrase

·         “the stem of the original design became unnecessary in everyday’s practice” – why?

·         “One of those is the McMinn cup (Waldemar Link  Hamburg, Germany).” – remove but leave the reference

·         “most orthopedic surgeons” – change to “current practice”

·         “after performing many prosthesis revision surgeries” – how many?

·         “McMinn cup”  - change to “cap with stem (McMinn cup (Waldemar Link  Hamburg, Germany))

·         “Revision acetabular cups and tantalum augments offer a widespread option. In case  of more important bone loss custom-made [10] titanium implants are not likely to become  a therapeutic standard due to their long manufacturing time and high cost. [11] In addition, it is anticipated that the new European Union Medical Device Regulation 2017/745 54 will make the use of custom-made implants difficult. [12]” – difficult to follow, should be rephrased

·         “we 59 developed our own method for these surgeries.” – what is the concept of this method?

·         “If we realized during revision that the bone defect is more important as suggested by  the CT scans, and the situation was suitable for using a McMinn cup” – rephrase

·         “metalworks” – change to “implants”

·         “The postoperative wound healing provided a good indicator  of a successful outcome.” – why?

·         “placed two marker screws into the 63 anterior part of the iliac crest. The postoperative wound healing provided a good indicator 64 of a successful outcome. A thin slice CT scan of the pelvis was then taken, allowing 3D  planning of the correct position and size of the McMinn's cup. Based on the two marker  screws and one more periacetabular bony part, we were designing an aiming device for  the insertion of the guide wire. A sterilizable aiming device was then 3D printed, and the 68 cup was inserted after about 3 weeks.” – this should be part of the Methods and not in the Introduction.

·         What is the concept of the surgical solution? What is the hypothesis? What are the objectives of this study?

Methods

·         “Engineers in the laboratory then initiated” – remove

·         “During the preop planning, we realized that despite the highly perfectioned soft- 88 ware, physicians’ supervision is necessary to ensure the requested position.” – why? What standard position was used?

·         How many patients and their demographic?

·         Inclusion criteria?

Results

·         Lines 150-170 and the table 1  should be in the Methods

·         There are no results in this report. Showing the postop ap xrays doesn’t mean a lot

To summarize – There is no hypothesis, no objectives, and no results suitable for interpretation. 

Not sufficient for a scientific text

Author Response

Dear Reviewer!

Thank you for your thorough review!

We have tried to improve the quality of our manuscript according to your suggestions and taking them into account.

Yours sincerely,
The authors

The following points should be addressed before the paper can be considered for publication

Abstract

  • What is the clinical problem? What is the concept of the surgical solution? What is the hypothesis? What are the results? – the abstract should be completely rewritten

The abstract  section has been rewritten. We believe that this is now much clearer.

Introduction

  • “so-called” – change to “a concept”

We have changed it.

  • “Developed during the  golden age of hip endoprosthetics,” – remove

We have removed it.

  • “The idea was based on” – change to “due to” and rephrase

We have corrected it.

  • “the stem of the original design became unnecessary in everyday’s practice” – why?

We have added a tagline to explain it.

  • “One of those is the McMinn cup (Waldemar Link  Hamburg, Germany).” – remove but leave the reference

It was removed.

  • “most orthopedic surgeons” – change to “current practice”

We changed it.

  • “after performing many prosthesis revision surgeries” – how many?

We have performed several hundred of prosthesis revision surgeries.

  • “McMinn cup”  - change to “cap with stem (McMinn cup (Waldemar Link  Hamburg, Germany))

We have changed those phrases to stemmed cup.

  • “Revision acetabular cups and tantalum augments offer a widespread option. In case  of more important bone loss custom-made [10] titanium implants are not likely to become  a therapeutic standard due to their long manufacturing time and high cost. [11] In addition, it is anticipated that the new European Union Medical Device Regulation 2017/745 54 will make the use of custom-made implants difficult. [12]” – difficult to follow, should be rephrased

We have rephrased it.

  • “we 59 developed our own method for these surgeries.” – what is the concept of this method?

We have written about it in the next two paragraphs.

  • “If we realized during revision that the bone defect is more important as suggested by  the CT scans, and the situation was suitable for using a McMinn cup” – rephrase

We have rephrased it.

  • “metalworks” – change to “implants”

We changed them.

  • “The postoperative wound healing provided a good indicator  of a successful outcome.” – why?

It allows early detection of the presence of bacteria.

  • “placed two marker screws into the 63 anterior part of the iliac crest. The postoperative wound healing provided a good indicator 64 of a successful outcome. A thin slice CT scan of the pelvis was then taken, allowing 3D  planning of the correct position and size of the McMinn's cup. Based on the two marker  screws and one more periacetabular bony part, we were designing an aiming device for  the insertion of the guide wire. A sterilizable aiming device was then 3D printed, and the 68 cup was inserted after about 3 weeks.” – this should be part of the Methods and not in the Introduction.

It was mentioned in the Introduction section and then it was explained in detail in the Materials section.

  • What is the concept of the surgical solution? What is the hypothesis? What are the objectives of this study?

The rewritten parts shed more light on these questions and our answers.

Methods

  • “Engineers in the laboratory then initiated” – remove

Removed.

  • “During the preop planning, we realized that despite the highly perfectioned soft- 88 ware, physicians’ supervision is necessary to ensure the requested position.” – why? What standard position was used?

We have added a sentence about it. „The main goal was the safe placement of the guiding wire from a predetermined point such a way, that the cup’s stem should not penetrate the surface of the cortical bone towards the pelvic cavity (linea terminalis).”

  • How many patients and their demographic?

We have added those datas.

  • Inclusion criteria?

All patients had to have major bone defects that could not have been resolved by conventional methods using a single implant.

Results

  • Lines 150-170 and the table 1  should be in the Methods

That part has also been reworded and restructured.

  • There are no results in this report. Showing the postop ap xrays doesn’t mean a lot

 We have written more about the results.

To summarize – There is no hypothesis, no objectives, and no results suitable for interpretation. 

We have improved the manual and made it easier to understand our objectives, criterion, challenges and results.

Comments on the Quality of English Language

Not sufficient for a scientific text

We have improved the language quality and solved the remaining issues.

Round 2

Reviewer 1 Report

I appreciate the detailed list of revisions that have been made to this scholarly article in response to feedback from peer reviewers. The comprehensive approach to enhancing each section of the paper—from improving the clarity and detail in the Materials and Methods to adding depth to the Results and Discussion—is commendable. Based on the extensive modifications I've made to address comments across multiple areas, including methodology, results, ethical considerations, and the quality of the English language, I believe the manuscript has been significantly improved and merits publication.

Author Response

Dear Reviewer,

Thank you for your review and appreciation!

Yours sincerely,
Lorand Csamer

Reviewer 3 Report

The following should be addressed be the paper could be considered for publication:

·         The English style is still deficient for scientific writing – it should be professionally edited

·         “foreign material” – clarify and use another term

·         “by the slow process of medical device 59 certification in Europe under the recently enforced regulation” – remove and rephrase

·         “we thought the single use of the stemmed cup, rather than 49 using auxiliary, could solve the problem of cup recovery” – This is probably your hypothesis, so it should be clearly defined as such

·         Line 66 should start with: “ According to our experience, we think that……

·         “Firstly, all implants were removed and were placed two” – remove and instead insert: “Accordingly we utilized a method based on ……” and rephrase by defining the general concept of the method that should include “to create a targeting device that allows the surgeon to  safely insert a stable guidewire in the right position, made of 3D printable, biocompatible  material.

·         “A thin slice CT scan of the pelvis was then taken, allowing 3D planning of the correct 71 position and size of the stemmed cup. Based on the two marker screws and one more  periacetabular bony part, an aiming device were designed for the insertion of the guide 73 wire. A sterilizable aiming device was then 3D printed, and the cup was inserted after  about 3 weeks.”  - should be part of the Methods and not in the Introduction

·         “Informed consent was obtained from all patients involved in this study. Between 231 February 2018 and July 2021, revision surgeries of large periacetabular defects (Paprosky 232 ≥ 2B) were performed with stemmed cup using the above-described method in a total of 233 17 patients. The age of the patients ranged from 35 to 77 years, the gender distribution 234 was 12 females and 5 males, and the follow-up period ranged from 10 months to 34 235 months. All patients belonged to the Caucasian race. See Table 1 for complete details. 236 The patients did not have any other conditions apart from musculoskeletal problems 237 that could have compromised the outcome of the surgeries. All patients had major bone 238 defects that could not have been resolved by conventional methods using a single implant. 239 The radiological examination allowed us to verify the close bone-to-implant contact 240 and the unchanged position of the implant during follow-up.” – should be part of Methods and not in Results

·         “For primary hip replacements, the Harris Hip Score (HHS) is used to assess hip function before and after surgery. In these severe revision cases it is not relevant. It can be 250 explained as follows: This group of patients had been severely limping for a significant 251 time prior to surgery. These patients are instructed to expect partial weight bearing after 252 surgery. Therefore, they are fundamentally different from patients in whom primary hip 253 arthroplasty is routinely performed for osteoarthritis.” - should be part of Methods and not in Results

·         What are the Results? What are the parameters that were investigated?

·         How do the results support or contradict the hypothesis of the study?

English style should be improved

Author Response

The following should be addressed be the paper could be considered for publication:

  • The English style is still deficient for scientific writing – it should be professionally edited

Although the paper has already been submitted to the MDPI language editing service before submission, and we have attempted to correct any errors made during the review in this version, we will use the service again if you find it necessary. However, we do not see this as fitting into the seven working days we have been given to make the corrections, so we have looked into all your other concerns and solutions, except this one, and we look forward to your reply!

  • “foreign material” – clarify and use another term

We have redefined and clarified the context of the problematic term.

  • “by the slow process of medical device 59 certification in Europe under the recently enforced regulation” – remove and rephrase

We have rephrased the sentence.

  • “we thought the single use of the stemmed cup, rather than 49 using auxiliary, could solve the problem of cup recovery” – This is probably your hypothesis, so it should be clearly defined as such

We have rewritten the sentence to define clearly our hypothesis.

  • Line 66 should start with: “ According to our experience, we think that……

The sentence has been rewritten to take account of your suggestion.

  • “Firstly, all implants were removed and were placed two” – remove and instead insert: “Accordingly we utilized a method based on ……” and rephrase by defining the general concept of the method that should include “to create a targeting device that allows the surgeon to  safely insert a stable guidewire in the right position, made of 3D printable, biocompatible  material.

On your recommendation, we have rewritten this paragraph.

  • “A thin slice CT scan of the pelvis was then taken, allowing 3D planning of the correct 71 position and size of the stemmed cup. Based on the two marker screws and one more  periacetabular bony part, an aiming device were designed for the insertion of the guide 73 wire. A sterilizable aiming device was then 3D printed, and the cup was inserted after  about 3 weeks.”  - should be part of the Methods and not in the Introduction

We have relocated this paragraph to the first part of Materials section.

  • “Informed consent was obtained from all patients involved in this study. Between 231 February 2018 and July 2021, revision surgeries of large periacetabular defects (Paprosky 232 ≥ 2B) were performed with stemmed cup using the above-described method in a total of 233 17 patients. The age of the patients ranged from 35 to 77 years, the gender distribution 234 was 12 females and 5 males, and the follow-up period ranged from 10 months to 34 235 months. All patients belonged to the Caucasian race. See Table 1 for complete details. 236 The patients did not have any other conditions apart from musculoskeletal problems 237 that could have compromised the outcome of the surgeries. All patients had major bone 238 defects that could not have been resolved by conventional methods using a single implant. 239 The radiological examination allowed us to verify the close bone-to-implant contact 240 and the unchanged position of the implant during follow-up.” – should be part of Methods and not in Results

Thank you for the suggestion! Based on the request of the first reviewer, patient demographics and surgical conditions were detailed in the Results section. We have just relocated these paragraphs to the end of Materials section.

  • “For primary hip replacements, the Harris Hip Score (HHS) is used to assess hip function before and after surgery. In these severe revision cases it is not relevant. It can be 250 explained as follows: This group of patients had been severely limping for a significant 251 time prior to surgery. These patients are instructed to expect partial weight bearing after 252 surgery. Therefore, they are fundamentally different from patients in whom primary hip 253 arthroplasty is routinely performed for osteoarthritis.” - should be part of Methods and not in Results

This explanation was also included in the section as instructed by the first reviewer. We have also relocated this paragraph to the end of Materials section.

  • What are the Results? What are the parameters that were investigated?

 We have added a paragraph to the beginning of the Results sections. With these lines we responded to the success criteria set out in the Surgical section. No misalignment occurred, as determined by visual inspection of postop X-ray images attached to Table 1. The HHS would have been a parameter accepted by the professionals, but it cannot be used for these severe cases, as described at the end of the Materials chapter.

  • How do the results support or contradict the hypothesis of the study?

Our results, summarized at the beginning of the Results section, support the hypothesis of hip revision surgery with a single stemmed cup without difficult situations.

Yours sincerely,

The authors